# Current Clinical Practice on the Management of Invasive Streptococcus Pyogenes Infections in Children: A Survey-Based Study

**DOI:** 10.3390/antibiotics14100970

**Published:** 2025-09-26

**Authors:** Maia De Luca, Costanza Tripiciano, Carmen D’Amore, Marta Luisa Ciofi Degli Atti, Lorenza Romani, Federica Pagano, Daniele Zama, Silvia Garazzino, Giangiacomo Nicolini, Samantha Bosis, Elena Chiappini, Claudia Colomba, Andrea Lo Vecchio

**Affiliations:** 1Infectious Disease Unit, Bambino Gesù Children’s Hospital, IRCCS, 00165 Rome, Italy; costanza.tripiciano@opbg.net (C.T.); lorenza.romani@opbg.net (L.R.); 2Epidemiology, Clinical Pathways and Clinical Risk, Medical Direction, Bambino Gesù Children’s Hospital, IRCCS, 00165 Rome, Italy; carmen.damore@opbg.net (C.D.); marta.ciofidegliatti@opbg.net (M.L.C.D.A.); 3PhD National Program in One Health Approaches to Infectious Diseases and Life Science Research Department of Public Health, Experimental and Forensic Medicine, University of Pavia, 27100 Pavia, Italy; federica.pagano@unina.it; 4Department of Translational Medical Sciences, University of Naples Federico II, 80125 Naples, Italy; andrea.lovecchio@unina.it; 5Department of Medical and Surgical Sciences, Alma Mater Studiorum, University of Bologna, 40126 Bologna, Italy; daniele.zama@aosp.bo.it; 6Pediatric Emergency Unit, IRCCS Azienda Ospedaliero-Universitaria di Bologna, 40138 Bologna, Italy; 7Department of Paediatrics, Infectious Diseases Unit, Regina Margherita Children’s Hospital, University of Turin, 10126 Turin, Italy; silvia.garazzino@unito.it; 8Pediatric Care Unit, San Martino General Hospital, 32100 Belluno, Italy; giangiacomo.nicolini@aulss2.veneto.it; 9Fondazione IRCCS Ca’ Granda Ospedale Maggiore Policlinico, 20122 Milano, Italy; samantha.bosis@unimi.it; 10Department of Health Sciences, Pediatric Infectious Diseases Unit Meyer Children’s University Hospital, IRCCS, University of Florence, 50121 Florence, Italy; elena.chiappini@unifi.it; 11Department of Health Promotion, Maternal and Infant Care, Internal Medicine and Medical Specialties, University of Palermo, 90133 Palermo, Italy; claudia.colomba@libero.it

**Keywords:** *Streptococcus pyogenes*, invasive infections, children, immunoglobulin, survey

## Abstract

**Background/Objectives**: *Streptococcus pyogenes* (Group A Streptococcus, GAS) is a major human pathogen that causes a wide spectrum of diseases. While mild infections like pharyngitis and impetigo are common, severe and invasive infections, though less frequent, pose significant health risks, particularly in children. In recent years, the re-emergence of hypervirulent GAS strains has heightened global concern. Nowadays, the absence of universally accepted guidelines compels clinicians to rely on a combination of clinical judgment, microbiological data and available evidence to manage these infections effectively. Our aim was to assess the current management of invasive GAS (iGAS) infections in Italy and the variability in therapeutic and preventive approaches. **Methods**: A web-based current clinical practice survey about invasive and severe GAS infections was designed according to the Checklist for Reporting of Survey Studies (CROSS) methodology and circulated among the members of the Italian Society of Pediatric Infectious Diseases (SITIP). **Results**: The survey reveals that while many practices are commonly shared among clinicians, particularly regarding first-line therapies (penicillin or ceftriaxone depending on the infection site), significant uncertainties remain, particularly about the use of combined antibiotic regimens and supportive treatments. The use of combined antibiotic regimens was considered appropriate as first-line therapy for STSS, NF and brain abscesses. Clindamycin was the preferred agent for combination with beta-lactam for most infections, except for brain abscesses, where linezolid was favored. However, there was disagreement regarding the optimal timing for de-escalation to beta-lactam monotherapy. Responses varied widely concerning the indications and dosages for IVIG, as well as the use of corticosteroids. **Conclusions**: Addressing the burden of invasive GAS (iGAS) infections in children requires enhanced surveillance, early recognition, prompt treatment and preventive strategies. Further work to increase surveillance, e.g., developing national registries, and to standardize the management of the disease, e.g., developing country-specific guidelines, is essential to build solid evidence on the most effective approaches.

## 1. Introduction

*Streptococcus pyogenes* (Group A Streptococcus, GAS) represents a public health concern due to potential life-threatening infections as well as non-infective complications which can follow GAS infections (e.g., acute rheumatic fever, post-streptococcal arthritis and glomerulonephritis). Invasive GAS (iGAS) infections commonly manifest as cellulitis, necrotizing fasciitis (NF), streptococcal toxic shock syndrome (STSS) and pneumonia. Other forms include osteomyelitis, meningitis, bacteremia, arthritis, myositis and endocarditis.

A 2005 global review estimated 163,000 deaths yearly worldwide caused by iGAS [1]. Several European studies performed in the last 5 years report a 2–4% case-fatality rate [2,3].

Moreover, in recent years, an increased incidence of iGAS infections has been described [4]. Van Kempen et al. recently described that risk factors for severe disease, defined as requiring admission to an intensive care unit and/or death, include post COVID-19 pandemic infection, necrotizing fasciitis, pulmonary involvement, meningitis or encephalitis and STSS, while multivariate analysis identified only STSS as a risk factor for mortality [5].

GAS remains universally sensitive to β-lactams. However, specific consideration must be made for invasive infections. Recommendations for NF and STSS suggest the combination of a β-lactam with clindamycin [6], which disrupts the production of protein virulence factors such as streptolysin S, protein M and pyrogenic exotoxins. Beyond its anti-toxic effects, clindamycin is used also to overcome the “Eagle effect” of beta-lactams in infections with high bacterial loads or slow-growing bacteria such as those in the stationary phase [7].

Alarmingly, between 2011 and 2019, the US Centers for Disease Control and Prevention (CDC) reported an increase in clindamycin resistance among GAS isolates from 8.9% to 23.8% [8]. Although current data are lacking and controversial, linezolid, that inhibits protein synthesis via the 50S ribosome subunit, is a promising alternative adjunctive antitoxin antibiotic [9].

Consensus on the indication and timing for adding a protein synthesis inhibitor antibiotic to β-lactams in the treatment of other iGAS infections remains limited, as the use of adjunctive therapies such as steroids and intravenous immunoglobulins (IVIGs).

Contacts of iGAS disease have an increased risk of contracting the infection [10,11]. However, a clear guideline for post-exposure prophylaxis is not available and indications vary worldwide.

In the absence of universally accepted guidelines, clinicians must rely on a combination of clinical judgment, microbiological data and expert opinions, which currently represents the highest level of evidence for the management of iGAS. The Italian Society of Pediatric Infectious Diseases (SITIP) conducted a survey among pediatric infectious disease (PID) specialists to summarize their opinion, assess the current practices and approaches and to investigate variations in the management of iGAS infections in children.

## 2. Results

Among the 73 specialists reached by email, 24 (32.8%) agreed to participate and completed the survey. The results are presented according to diseases’ localization (Figure 1).

Sepsis

Over 75% of the respondents (18/24) agreed or strongly agreed with the use of penicillin or ampicillin as first-line therapies. Only 54.1% (13/24) agreed or strongly agreed with ceftriaxone in this context, while 75% (18/24) disagreed or strongly disagreed with the use of vancomycin. Furthermore, less than half of respondents (41.6%, 10/24) would prescribe combined antibiotic therapy (beta-lactam + other agent with activity on GAS) as first-line therapy, while 50% (12/24) would reserve it for patients not responding to beta-lactams after 48–72 h of treatment.

Regarding combination therapy, 95.8% (23/24) agreed or strongly agreed with adding clindamycin to beta-lactams. The options of linezolid, tedizolid and vancomycin as combination therapy in association to beta-lactams did not reach a >75% agreement rate. Moreover, 54.2% (13/24) agreed or strongly agreed to continue combined antibiotic therapy until clinical and hemodynamic stability. Regarding the duration of antibiotic therapy, 45.8% (11/24) agreed or strongly agreed that 7–10 days or more should be completed if a complicating deep-site infection was detected.

Streptococcal toxic shock syndrome

Penicillin, ampicillin and ceftriaxone were considered appropriate (“Agree” + “Strongly agree”) as first-line therapies of STSS due to GAS by 66.6% (16/24), 75% (18/24) and 70.7% (17/24) of respondents, respectively. A total of 79.2% (19/24) of the specialists agreed or strongly agreed with early use of combined antibiotic therapy (beta-lactam + other agent with activity on GAS). In these cases, 95.8% (23/24) of the specialists agreed or strongly agreed with adding clindamycin to beta-lactams, 66.6% (16/24) with adding linezolid. Regarding the best time to discontinue combination therapy by shifting to monotherapy, no option exceeded 50% agreement. A total of 83.3% (20/24) of the respondents would continue the antibiotic treatment for at least 14 days, or longer if a complicating deep-site infection was detected.

Necrotizing fasciitis

The specialists agreed or strongly agreed with the use of penicillin, ampicillin or ceftriaxone as first-line therapy of NF caused by GAS in 75% (18/24), 75% (18/24) and 66.6% (16/24) of cases, respectively. A total of 100% of respondents would use combined antibiotic therapy from the beginning of the treatment. In these cases, all the specialists agreed or strongly agreed with adding clindamycin to beta-lactam monotherapy and 70.8% (17/24) with adding linezolid. A total of 50% (12/24) of the panelists agreed or strongly agreed with the continuation of combined antibiotic therapy until the end of the whole treatment and 91.7% (22/24) agreed or strongly agreed with the following response option: “length of therapy depends on the clinical course and the adequacy of surgical debridement; usually therapy is continued for 14 days from the last positive culture obtained during surgical debridement”.

Pneumonia

More than 75% of the respondents agreed or strongly agreed with the use of penicillin or ampicillin as first-line therapies for GAS pneumonia. A total of 70.8% (17/24) of respondents agreed or strongly agreed with the use of ceftriaxone, while 74.9% (18/24) disagreed or strongly disagreed with the use of vancomycin. The specialists assigned mixed Likert Scale scores on the use of combination therapy as first-line therapy, but they agreed or strongly agreed to add a second agent with activity on GAS in addition to beta-lactam in non-responders to treatment after 48–72 h (91.6%, 22/24), in patients with evidence of necrosis at chest XR/CT scan (100%) and in patients with pleural effusion >2/3 of the hemithorax (83.2%, 20/24). When a second agent was deemed necessary, 83.2% (20/24) and 79.1% (19/24) of respondents agreed or strongly agreed with the use of clindamycin or linezolid, respectively, while 62.4% (15/24) disagreed or strongly disagreed with the use of vancomycin. None of the options regarding the best time to stop combination therapy by switching to beta-lactam monotherapy exceeded 50%.

Otomastoiditis

Specialists agreed or strongly agreed with the use of ceftriaxone as first-line therapy for the treatment of otomastoiditis caused by GAS in 87.4% (21/24), while penicillin and ampicillin in 70.7% (17/24) and in 58.2% (14/24) of cases, respectively. A total of 79.2% (19/24) of the respondents would add a second agent with activity on GAS in addition to the beta-lactam non-responders to treatment after 48–72 h. In this setting, 83.2% (20/24) agreed or strongly agreed with the use of clindamycin, while only 70.7% (17/24) and 54% (13/24) with linezolid and vancomycin. None of the options regarding the best time to stop combination therapy by switching to beta-lactam monotherapy exceeded 50%.

Meningitis and brain abscess

Ceftriaxone was the only response option to achieve >75% agreement for the first-line treatment of both meningitis and brain abscess. Ampicillin achieved a favorable response in 54% (13/24) of cases for meningitis and only 41.7% (10/24) of cases for brain abscess, while vancomycin achieved 33.3% (8/24) and 58.2% (14/24), respectively. The specialists panel agreed or strongly agreed with using combined antibiotic therapy as first-line therapy versus reserving it for non-responders after 48–72 h in 50% (12/24) of cases for meningitis and 87.4% (21/24) of cases for brain abscess. When considered appropriate, respondents agreed or strongly agreed with the use of linezolid as a second agent in addition to beta-lactam in 70.7% (17/24) for meningitis and 75% (18/24) of cases for brain abscess; clindamycin did not exceed 50% of favorable responses, while vancomycin was considered appropriate only in 58.2% (14/24) of cases for meningitis and 66.6% (16/24) of cases for brain abscess. None of the options regarding the best time to stop combination therapy by switching to beta-lactam monotherapy exceeded 50% for meningitis; a total of 79.2% (19/24) of respondents agreed or strongly agreed with switching to beta-lactam monotherapy in patients with brain abscess when the MRI showed a significant lesion reduction.

Septic arthritis and osteomyelitis

Penicillin, ampicillin and ceftriaxone were considered appropriate (“Agree” + “Strongly agree”) as first-line therapies of septic arthritis/osteomyelitis due to GAS by 58.3% (14/24), 70.8% (17/24) and 70.8% (17/24) of respondents, respectively. The specialists disagreed or strongly disagreed with the use of vancomycin as first-line monotherapy in 54.2% (13/24) of cases. A total of 70.8% (17/24) of the respondents would add a second antibiotic with activity on GAS in addition to the beta-lactam only in non-responders after 48–72 h; in this context, clindamycin received favorable responses in 91.7% (22/24) of cases, while linezolid and vancomycin in 62.5% (15/24) and 54.2% (13/24) of cases, respectively. None of the response options regarding the best time to stop combination therapy by switching to beta-lactam monotherapy exceeded 50%.

Use of intravenous immunoglobulins

Respondents did not exceed 50% agreement or disagreement in the use of IVIG for the treatment of iGAS infections. However, 50% or more of the specialists agreed or strongly agreed that IVIG should be reserved for hemodynamically unstable patients (50%, 12/24), those admitted to ICU (58.3%, 14/24) or affected by STSS (70.8%, 17/24) or NF (50%, 12/24). A total of 58.3% (14/24) of the specialists preferred the dosage of 1 g/kg on the first day followed by 0.5 g/kg on the next day. [Figure 2A].

Use of steroids

A total of 54.1% (13/24) of the respondents stated that they would never use steroids for the treatment of otomastoiditis caused by GAS; in particular, 70.8% (17/24) disagreed or strongly disagreed with the use of steroids as first-line therapy in this context. In patients with meningitis, 66.6% (16/24) of the respondents disagreed or strongly disagreed with the sentence “I would never use steroids”; however, only 45.8% (11/24) and 16.6% (4/24), respectively, agreed or strongly agreed with the use of steroids as first-line therapy or as a treatment reserved for patients who do not respond to beta-lactams after 48–72 h of treatment. A total of 75% (18/24) of the specialists agreed or strongly agreed with the use of steroids in patients with evidence of edema at the brain MRI. Similar rates were also obtained for questions regarding the management of brain abscesses. The data are summarized in [Figure 2B].

Prophylaxis of close contacts

In our survey, close contacts were defined as individuals who had prolonged contact with the case in a household-type setting during the seven days before the diagnosis of an iGAS infection and up to 24 h after the initiation of appropriate antimicrobial therapy in the index case (e.g., overnight stay in the same household including extended household if the case has stayed at another household, pupils in the same dormitory, intimate partners or university students sharing a kitchen in a hall of residence). For a care home, close contact is defined as someone sharing a bedroom. The panel agreed or strongly agreed with this definition by 70.8% (17/24).

The specialist panel was then asked to identify the categories of high-risk contacts who might benefit from antibiotic prophylaxis. The percentages of agreement in administering prophylaxis to the following categories of high-risk close contacts were as follows: neonates (62.5%, 15/24), >65 years (70.8%, 17/24), immunocompromised (75%, 18/24), recent surgery (37.5%, 9/24), household contacts regardless of age (20.8%, 5/24), household contacts if two or more confirmed or probable cases occur in the same family unit (54.2%, 13/24), close contacts if two or more confirmed or probable iGAS cases occur in a community within one month (50%, 12/24), pregnant women > or =37 weeks (54.2%, 13/24), post-partum women < or =28 days (58.3%, 14/24) and individuals with chickenpox active lesions within seven days prior to the diagnosis of an iGAS infection in the index case or within 48 h after commencing antibiotics by the iGAS case if exposure is ongoing (70.8%, 17/24). Only 33.3% (8/24) of the respondents agreed or strongly agreed to reserve prophylaxis for high-risk close contacts with positive pharyngeal swabs for GAS. Regarding appropriate timing for prophylaxis, the following definition was proposed to the specialists: “For maximum benefit, chemoprophylaxis should be administered as soon as possible (within 24 h, and preferably the same day) after eligible contacts are identified and not beyond 10 days after iGAS diagnosis in the index case. Advise GPs to maintain a low threshold of suspicion for 30 days in all close contacts”. A total of 83.3% (20/24) of the specialists agreed or strongly agreed with this definition.

A list of antibiotics was proposed as possible options for chemoprophylaxis. The agreement rates were as follows: penicillin (33.3%, 8/24), amoxicillin (87.5%, 21/24) and cephalexin (58.3%, 14/24). In case of a beta-lactam allergy, the panel agreed or strongly agreed with the use of clarithromycin, azithromycin or clindamycin in 62.5% (15/24), 58.3% (14/24) and 62.5% (15/24) of cases, respectively [Figure 3].

## 3. Discussion

We report the results of a survey regarding the management of iGAS infections in pediatric patients conducted among 24 PID specialists, sponsored by the SITIP.

More than 75% of the specialists would prescribe beta-lactam as the first-line therapy for the treatment of an iGAS infection with differences depending on the location (penicillin or ampicillin for sepsis, STSS, NF and pneumonia; ceftriaxone for otomastoiditis, meningitis and brain abscesses; and ampicillin and ceftriaxone were the preferred options for arthritis and osteomyelitis). This is in line with the high susceptibility of *GAS* to beta-lactams, as reported in the European literature [12]. However, over the last 20 years, increased MICs to penicillin and cephalosporins have been reported in Asian, African and Central American countries, caused by mutations within PBPs [13]; thus, continuous surveillance of the GAS population should be of critical interest for public health.

Combined antibiotic therapy as first-line therapy was considered appropriate for the treatment of STSS, NF and brain abscesses, while in the other conditions the specialists would reserve combination therapy for non-responders to treatment after 48–72 h. Clindamycin was the preferred agent for combination with beta-lactam for most infections; linezolid was preferred for brain abscesses only. None of the options regarding the best time to stop combination therapy by switching to beta-lactam monotherapy exceeded 75%; only for brain abscesses, 79.2% of respondents would switch to monotherapy when MRI shows a significant lesion reduction. The use of an adjunctive agent with anti-toxic effects (i.e., clindamycin or linezolid) has been demonstrated to reduce mortality in iGAS infections by in vitro and animal models, as well as multiple observational studies [14]; however, randomized clinical trials are missing. Coyle E. et al. used an in vitro model to show that the addition of linezolid or clindamycin to penicillin exhibited a significantly lower SPE A release than penicillin alone [15]. Although Heil et al. demonstrated that there was no difference in the reduction in SOFA scores between the baseline and 72 h among patients treated with clindamycin vs. linezolid [16], clindamycin resistance in GAS has rapidly increased in the US since the mid-2010s. In Europe, a rising clindamycin resistance from 7.8% to 28% has been recently noted in Ireland [17]. Linezolid offers potential advantages including universal susceptibility among GAS, more favorable side effect profile, CNS penetration and concomitant MRSA coverage. Further studies comparing clindamycin to linezolid are needed. However, it is important to underline that its use in pediatrics is still off-label in several European countries, including Italy.

Response rates on the indication and dosages of IVIG and steroid use varied widely among participants. In 2003, Darenberg et al. found a 3.6-fold higher mortality rate in the placebo group compared to patients receiving IVIG [18]. A significant decrease in the SOFA score on days two and three was also noted in the IVIG group. However, no data comparing different regimens of IVIG were available and a high drop-out rate was registered throughout the study. Robust pediatric-specific data on IVIG use in iGAS infections are missing. Concerning the use of steroids, the potential benefits for the treatment of non-pneumococcal and non-*Haemophilus* bacterial meningitis are still debated [19]. Fewer than half of our specialists would use steroids as first-line therapy for patients with meningitis or brain abscesses, while most of them agreed with the use of steroids in patients with MRI-confirmed edema, aiming to reduce mortality and hearing loss. To date, cases of iGAS meningitis reported in the literature are limited. In 2025 Di Meglio L et al. [20] reviewed all 57 cases described to date, reporting that steroid use in iGAS meningitis did not differ statistically between patients with an uneventful clinical course and those with complications. However, in many instances, steroids were introduced only after MRI evidence of brain edema, rather than immediately after bacterial detection, making it impossible to draw conclusions on the efficacy of early steroid use. A total of 70.8% of the specialists agreed with the definition of “close contacts” of the index case proposed by the UK guidelines. The rates of agreement with the pre-defined categories of “high-risk contacts” that could benefit from antibiotic prophylaxis were highly variable, with only “immunocompromised” approaching 75%. Moreover, only 37.5% and 20.8% of the respondents, respectively, would offer prophylaxis to patients with recent surgery or co-habitants with the index case. The low rates of agreement among respondents are likely due to the fact that the currently available evidence on this topic is limited and largely based on studies with weak designs and small sample sizes. Additionally, existing guidelines provide different recommendations depending on the country.

For example, the UK guidelines published in December 2022 recommend offering chemoprophylaxis to the following categories of individuals: pregnant women from ≥37 weeks’ gestation, neonates and women within the first 28 days postpartum (regardless of whether either was the index case), household contacts aged ≥75 years and individuals who develop chickenpox with active lesions within seven days prior to the diagnosis of an iGAS infection in the index case or within 48 h after the index case starts antibiotics, if exposure is ongoing.

In contrast, the American Academy of Pediatrics (AAP), in the 2021 edition of the *Red Book*, limits its recommendations to household contacts aged ≥65 years or members of other high-risk populations (e.g., individuals with HIV infection, varicella or diabetes mellitus) [21].

In the Netherlands, before January 2023, antibiotic prophylaxis was only recommended for household contacts of individuals with specific severe iGAS presentations, such as necrotizing fasciitis or streptococcal toxic shock syndrome (STSS). After 2023, the policy was expanded to include household contacts of all iGAS cases [22]. Currently, there are no clear national recommendations in Italy.

A high rate of consensus on the proposed timing for prophylaxis administration and choice of amoxicillin was found instead.

Study limitations include potential biases inherent the survey study design. The response rate of our survey was 32.8%, which is relatively low; however, this figure is comparable to other surveys conducted among busy medical specialists and should be interpreted in light of the demanding clinical workload of this professional group. As such, some degree of non-response bias cannot be excluded. In addition, self-selection bias may have occurred; ID specialists with a stronger interest in the survey topic may have been more likely to participate compared to those who did not. A temporal bias is also possible, since the survey was conducted shortly after the European outbreak of iGAS infections. Furthermore, several aspects could not be explored in order to limit the number of questions in the survey and optimize the adherence (e.g., antimicrobial treatment optimization, posology and treatment duration). Finally, our results should not be taken as statistically representative of all pediatricians, but rather as a structured expert consensus to be interpreted as hypothesis-generating and as a starting point for larger, multicenter studies and guideline development.

## 4. Materials and Methods

### 4.1. Study Design, Data Collection and Analysis

A comprehensive literature review was conducted to identify well-established areas and key uncertainties regarding iGAS management [23]. Based on this, two researchers developed clinical questions using the PICO framework (Patient/Population/Problem, Intervention, Comparison, Outcome). Each component of the PICO questions was then carefully translated into clear, specific survey items designed to be easily understood and answered by the target audience, minimizing ambiguity.

To ensure methodological rigor and transparency, the CROSS checklist was applied to address essential survey elements [24]. Following this, the survey draft was submitted to the SITIP Council for review, feedback and approval.

The survey consisted of 62 questions grouped into 11 main topics: 9 regarding the treatment of invasive infections (bacteremia, STSS, NF, pneumonia, otomastoiditis, meningitis, brain abscess, septic arthritis and osteomyelitis) and 2 about the use of immunoglobulins and chemoprophylaxis of contacts. The participants were asked to provide an answer to multiple choices questions, or alternatively, to express their level of agreement by using a 5-point Likert scale (from 1 = Strongly Disagree to 5 = Strongly Agree). The complete questionnaire is reported as Appendix A.

The survey was converted into Google Forms and then circulated by e-mail among PID specialists with a minimum of 5 years of experience in the field and the following additional requirements: being members of the Italian and/or European Societies of PID and working in tertiary care institutions caring for children. The participants received reminders to complete the survey in December 2023, January and February 2024.

The questionnaire was considered eligible for data analysis only if completely answered within the deadline. Data were extracted into an Excel file and anonymized. Data were analyzed by expressing categorical data as counts and proportions and then visualized using R and Rstudio (© Posit software, release 2024.12.0 + 467). A pre-final manuscript was created and submitted to all participants for evaluation and final comments.

### 4.2. Definitions

For the scope of this study, in accordance with the literature, the following definitions were adopted [25]:▯Invasive group A Streptococcus infection: detection of *S. pyogenes* by culture or accredited molecular methods (such as PCR), from a normally sterile body site, such as blood, cerebrospinal fluid, joint aspirate, pericardial-peritoneal-pleural fluids, bone, endometrium, deep tissue or deep abscess at operation.▯Severe group A Streptococcus infection: isolation of *S. pyogenes* from a normally non-sterile site such as the throat, sputum, vagina or a wound in combination with severe clinical presentation, such as STSS, NF, pneumonia, septic arthritis, meningitis, peritonitis, osteomyelitis, myositis and puerperal sepsis.

First-line therapy was defined as the antibiotic agent(s) selected after the confirmation of etiological diagnosis of GAS infection, while the antibiotic susceptibility results were still unknown.

### 4.3. Ethical Statement

The survey was addressed exclusively to healthcare professionals. The subjects involved were adequately informed about the aims and procedures of the research, so that they were aware and free to express their adherence to the study protocol and withdraw from it at any time without any penalty. Before answering the questionnaire, participants were also informed about the anonymous data sharing, analysis and publication of the data. The questionnaire regarded exclusively clinical practices, while no personally identifiable information or data about patients were collected. Answers were registered in anonymous form. This study was conducted in accordance with the Declaration of Helsinki and the Italian Personal Data Protection Code (Legislative Decree No. 196 of 30 June 2003) and subsequent amendments and additions. The authors did not declare potentially relevant conflicts of interest.

## 5. Conclusions

The management of iGAS infections remains a global challenge. In the absence of international guidelines, some common practices exist, but many uncertainties persist, particularly regarding combined antibiotic therapies and supportive treatments. Further work to standardize the management of the disease is essential to build solid evidence on the most effective approaches. We highlight potential areas for future scientific studies aimed at developing country-specific guidelines and harmonizing treatment strategies worldwide.

### Future Directions

Enhanced surveillance of local GAS strains to monitor antibiotic susceptibility trends;Clinical trials to assess outcomes in patients treated with combination therapy (including anti-toxin antibiotics) versus monotherapy;Comparative studies on the timing of anti-toxin antibiotic initiation (first-line versus delayed use at 48–72 h) across various infectious sites;Evaluation of the efficacy, safety and cost-effectiveness of clindamycin versus linezolid as anti-toxin agents;Research into the role of IVIG in different clinical scenarios (i.e., hemodynamic instability, ICU admission, NF or STSS), including optimal dosing regimens;Investigation of corticosteroid use in iGAS meningitis, analogous to pneumococcal meningitis treatment strategies;Development of comprehensive clinical guidelines to support the management of iGAS infections and strengthen outbreak preparedness among ID specialists.

## Figures and Tables

**Figure 1 antibiotics-14-00970-f001:**
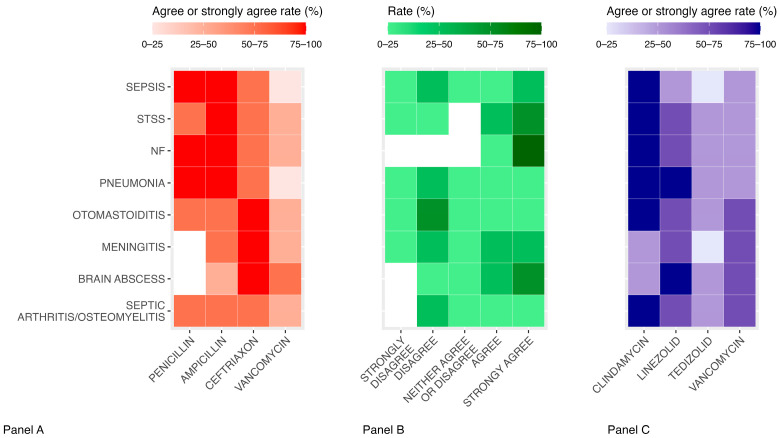
Agreement on first-line therapy options for iGAS infections: first-line antimicrobial agent (Panel (**A**)), combination therapy as first-line (Panel (**B**)), first choice for combination with beta-lactams (Panel (**C**)). Abbreviations: NF = necrotizing fasciitis, STSS = streptococcal toxic shock syndrome.

**Figure 2 antibiotics-14-00970-f002:**
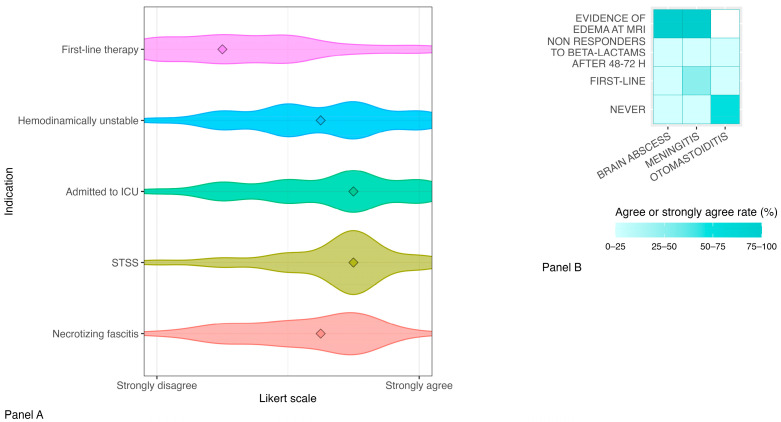
Supportive therapy for iGAS infections: density-based distribution of the Likert scale options for indications for intravenous immunoglobulins (Panel (**A**)) and agreement on indication for steroid prescription (Panel (**B**)). Abbreviations and symbols: ICU = intensive care unit, STSS = streptococcal toxic shock syndrome, MRI = magnetic resonance imaging, ◊ = median value.

**Figure 3 antibiotics-14-00970-f003:**
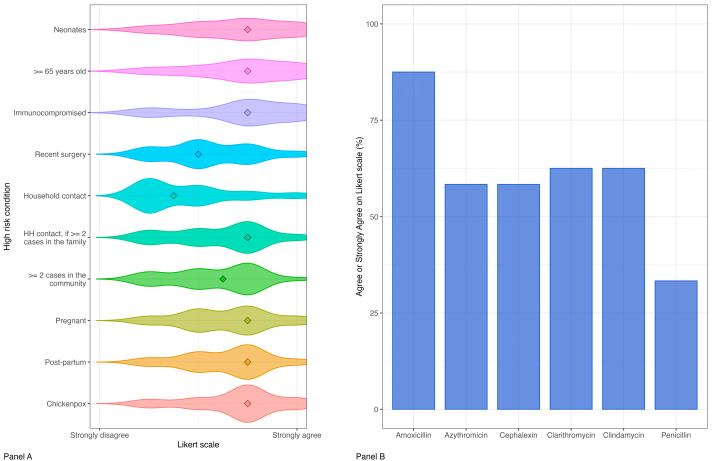
Prophylaxis of close contacts: density-based distribution of the Likert scale options for antibiotic prophylaxis for iGAS close contacts (Panel (**A**)) and choice of antibiotic for prophylaxis (Panel (**B**)). Abbreviations and symbols: HH = household, ◊ = median value.

## Data Availability

The dataset used for the study is made available from the corresponding author on reasonable request.

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
