# Peer review of "Current Clinical Practice on the Management of Invasive Streptococcus Pyogenes Infections in Children: A Survey-Based Study"

_antibiotics, 2025, doi:10.3390/antibiotics14100970_

Round 1
Reviewer 1 Report
Comments and Suggestions for Authors
File attached.

Author Response
Comments 1: Basing on your abstract it is not identified the most critical step: implementing a national iGAS registry to improve surveillance, or convening a consensus panel to establish
Italian-specific guidelines for therapy and prophylaxis.
Response 1: We have accepted the reviewer’s suggestion by adding a mention in the Abstract conclusions to the importance of developing national iGAS registry and country specific guidelines (Lines 52-54).
Comments 2: Line 42/43; needs clarification., significant uncertainties remain, particularly about the use of combined antibiotic regimens and supportive treatments (i.e., intravenous immunoglobulin and steroid use).
Response 2: We have further detailed the Abstract results, particularly regarding the disagreement areas among the respondents to the survey (Lines 44-49).
Comments 3: Only 24 specialists, what steps were taken to assess the potential for non-response bias, and how do you suggest clinicians interpret these results in light of this limited sample size?
Response 3: We thank the reviewer for raising this important point. We acknowledge that our sample size is limited and that our response rate (32.8%) is relatively low. However, this figure is in line with many surveys conducted among busy medical specialists, where response rates frequently range between 30–40%. Importantly, our respondents were experienced pediatric infectious disease specialists working in tertiary care centers, and we believe their answers still provide valuable insights into current clinical practice. Nevertheless, we recognize that non-response bias cannot be excluded, and we have highlighted this as a limitation in the revised manuscript (Lines 339-342).
Regarding interpretation, our results should not be taken as statistically representative of all pediatricians, but rather as a structured expert consensus reflecting the current variability and challenges in the management of severe GAS infections in children. We have emphasized this point in the discussion, highlighting that our findings should be interpreted as hypothesis-generating and as a starting point for larger, multicenter studies and guideline development (Lines 349-352).
Comments 4: The survey sample was limited to highly specialized pediatric infectious disease (PID) physicians working in tertiary care centres in Italy/Europe. So, could it be generalized.
Response 4: Thank you for your observation. Our findings cannot be directly generalized beyond the surveyed context of highly specialized PID physicians in Italy/Europe. Given the diversity of epidemiological settings worldwide (both for GAS infections and their non-infectious complications), as well as the recent regional variations in the susceptibility profiles of GAS strains, we believe that the development of any potential clinical recommendations should be based on the assessment of local epidemiology and current practices.
Comments 5: Only 54.1% agreement, What do you believe is driving this lack of consensus, and what does it suggest about the current state of evidence-based guidelines.
Response 5: It is not possible to answer this question. Please indicate which lines in the manuscript you are referring to.
Comments 6: Elaborate on what specific clinical scenarios or patient factors your respondents believe would justify steroid use in bacterial meningitis?
Response 6: We thank the reviewer for this valuable comment. As reported in the manuscript, while fewer than 50% of respondents would use steroids as first-line therapy - an approach typically recommended for meningitis of other etiologies such as Streptococcus pneumoniae-, 75% of respondents indicated that they would start corticosteroids in cases of bacterial meningitis or brain abscess associated with radiological signs of cerebral edema. Our survey did not include more detailed clinical scenarios or patient subgroups, and therefore we are unable to provide further stratification of the factors influencing this decision. Nonetheless, as in other forms of bacterial meningitis, the rationale for steroid use in the presence of cerebral edema would be to reduce the risk of mortality and neurologic sequelae, including hearing loss (Lines 304-310).
Comments 7: What factors do you believe contributed to the lack of agreement for the various categories line 283-87.
The lack of agreement among respondents is likely due to the fact that the currently available evidence on this topic is limited and largely based on studies with weak designs and small sample sizes. Additionally, existing guidelines provide different recommendations depending on the country.
For example, the UK guidelines published in December 2022 recommend offering chemoprophylaxis to the following categories of individuals: pregnant women from ≥37 weeks’ gestation, neonates, and women within the first 28 days postpartum (regardless of whether either was the index case), household contacts aged ≥75 years, and individuals who develop chickenpox with active lesions within 7 days prior to the diagnosis of iGAS infection in the index case or within 48 hours after the index case starts antibiotics, if exposure is ongoing.
In contrast, the American Academy of Pediatrics (AAP), in the 2021 edition of the Red Book, limits its recommendations to household contacts aged ≥65 years or members of other high-risk populations (e.g., individuals with HIV infection, varicella, or diabetes mellitus).
In the Netherlands, before January 2023, antibiotic prophylaxis was only recommended for household contacts of individuals with specific severe iGAS presentations, such as necrotizing fasciitis or streptococcal toxic shock syndrome (STSS). After 2023, the policy was expanded to include household contacts of all iGAS cases.
Currently, there are no clear national recommendations in Italy.
We added these considerations to the discussion (Lines 316-335).
Comments 8: Very small sample size.
Response 8: See response 3
Reviewer 2 Report
Comments and Suggestions for Authors
Congratulations to the authors.
I suggest minimal additions.
They describe very well the confusion one feels in the absence of a proper guideline. The last one, but only for soft tissue infections, is a 2014 IDSA guideline, which by the way still holds up today!
In the primary treatment, ceftriaxone is acceptable due to the uncertainty and the serious condition, but they did not ask whether they would switch to penicillin if GAS was confirmed, because I think that is a key question.
If the diagnosis is clear, then I also think penicillin is the first choice, perhaps ampicillin is an acceptable alternative.
The combination with clindamycin or linezolid is necessary not only to inhibit the production of toxic proteins, but also because of the Eagle effect. If there are too many bacteria in a given area, they will not multiply (no false synthesis), so the antibiotics that act on them will not be effective. That is why it must be given immediately in necrotizing fasciitis, but it does not hurt in the others either. The combination is usually recommended for 3 days, but there is no consensus. I think it is determined by the clinic.
Meningitis and brain abscess are a different matter. There, the choice is primarily determined by the passage through the blood-brain barrier, but in principle penicillin could also be given, but in high doses.
There is no mention of doses, although that would also be important.
How long do we give it?
Str. pyogenes can cause rheumatic fever wherever it causes disease. Of course, you need the right strain for this and the ones currently circulating in Europe do not cause it, but despite this, the 10-day penicillin treatment is a central dogma that has not yet been overturned, although there are articles that consider a 6-day treatment sufficient, and there are even those that say nothing happens if we do not treat a follicular tonsillitis. However, this 7-14-21 etc. is not up-to-date today because treatment should be stopped based on the clinic. They could comment on this!
Immunoglobulin is really not clear in its indication and dosage.
It never occurs to us to give steroids, except if intensive treatment makes it necessary (hypotension uncontrollable with vasoactive drugs).
You refer to the SOFA, which I think is not accepted in childhood, but there is already the Canadian score, which no one uses, at least I haven't read much about it yet. They could comment on that too!
The manuscript has 13 authors, half of the number who answered questions, why don't they evaluate this or that answer? What do they consider correct?
They could emphasize even more that further work on standardizing the treatment of the disease is essential.
Author Response
|
Congratulations to the authors. I suggest minimal additions. They describe very well the confusion one feels in the absence of a proper guideline. The last one, but only for soft tissue infections, is a 2014 IDSA guideline, which by the way still holds up today! |
Comments 1: In the primary treatment, ceftriaxone is acceptable due to the uncertainty and the serious condition, but they did not ask whether they would switch to penicillin if GAS was confirmed, because I think that is a key question. If the diagnosis is clear, then I also think penicillin is the first choice, perhaps ampicillin is an acceptable alternative.
Response 1: Thank you for your insightful comment. Descalation therapy is a core element of the antimicrobial stewardship programs. However, in order to optimize the participation in our 62-question survey, we had to focus on specific topics. We prioritized questions regarding initial management, as literature suggests that first-line therapy is a major determinant of clinical outcomes. Other important aspects, such as de-escalation to penicillin, antibiotic dosing, and treatment duration, were not fully explored in this survey. We have added this consideration to the study limitations (Lines 347-351).
We agree with the reviewer’s point regarding penicillin. While our survey did not explicitly address use of penicillin, we think that indications for penicillin use as first-line therapy in iGAS infections could reasonably be inferred from data on ampicillin.
Comments 2: The combination with clindamycin or linezolid is necessary not only to inhibit the production of toxic proteins, but also because of the Eagle effect. If there are too many bacteria in a given area, they will not multiply (no false synthesis), so the antibiotics that act on them will not be effective. That is why it must be given immediately in necrotizing fasciitis, but it does not hurt in the others either. The combination is usually recommended for 3 days, but there is no consensus. I think it is determined by the clinic.
Response 2: We thank you for raising this important point. We added a sentence in the text to explain the role of the Eagle effect (Lines 79-81).
Comments 3: Meningitis and brain abscess are a different matter. There, the choice is primarily determined by the passage through the blood-brain barrier, but in principle penicillin could also be given, but in high doses. There is no mention of doses, although that would also be important. Response 3: We thank the Reviewer for this valuable comment. We acknowledge that dosage is a key aspect in the management of invasive infections. However, in our survey we deliberately chose not to address antimicrobial dosing, in order to keep the questionnaire concise and facilitate its completion by physicians. Moreover, current literature does not report pathogen-specific dosage adjustments for invasive GAS infections. Therefore, we believe that clinicians can rely on the standard dosing recommendations available in the literature, which are usually tailored to the site of infection.
Comments 4: How long do we give it? Str. pyogenes can cause rheumatic fever wherever it causes disease. Of course, you need the right strain for this and the ones currently circulating in Europe do not cause it, but despite this, the 10-day penicillin treatment is a central dogma that has not yet been overturned, although there are articles that consider a 6-day treatment sufficient, and there are even those that say nothing happens if we do not treat a follicular tonsillitis. However, this 7-14-21 etc. is not up-to-date today because treatment should be stopped based on the clinic. They could comment on this!
Response 4: We thank the Reviewer for raising this important point. We agree that the optimal duration of antibiotic therapy, particularly for pharyngitis, is still a matter of debate. Even international guidelines report different recommendations, reflecting the lack of consensus on this issue. For this reason, we chose not to explore in detail the duration of treatment in the survey, also considering that for several invasive sites of infection (e.g., pneumonia, otomastoiditis, brain localizations) the length of therapy is largely influenced by the clinical course and patient’s response rather than by fixed schemes. We therefore limited our discussion on treatment duration only to those clinical settings where the literature provides relatively well-defined, pathogen-specific indications (NF and STSS).
Comments 5: Immunoglobulin is really not clear in its indication and dosage. It never occurs to us to give steroids, except if intensive treatment makes it necessary (hypotension uncontrollable with vasoactive drugs).
Response 5: Thank you for sharing your experience. We agree that the indications for immunoglobulin use remain unclear in the literature, partly because immunoglobulins are often co-administered with anti-toxin agents, making it difficult to isolate the effect of each treatment on mortality reduction. Similarly, the use of steroids lacks strong evidence. In our survey, we specifically asked specialists about steroid use only in conditions where the indication might be pathogen-specific—such as meningitis, where Pneumococcus has well-established recommendations.
Cases of iGAS meningitis reported in the literature are limited (57 cases up to December 2024, see Di Meglio et al., Microorganisms 2025), which makes drawing firm conclusions challenging. In our center, we admitted a young patient with meningitis who developed cerebral edema and died within 48 hours, raising the question of whether early steroid administration could have altered the outcome.
However, survey respondents generally do not support steroid use as first-line therapy but would reserve it for patients with MRI-confirmed edema, with the goal of reducing mortality and hearing loss.
We modified the text according to these considerations(Lines 304-310).
Comments 6: You refer to the SOFA, which I think is not accepted in childhood, but there is already the Canadian score, which no one uses, at least I haven't read much about it yet. They could comment on that too!
Response 6: The SOFA score was cited in reference to studies [13 and 15] that describe comparative effectiveness of different treatment strategies in iGAS in adult patients. While we acknowledge that the SOFA score is not validated for use in pediatric populations, we chose to reference these studies due to the lack of robust, pediatric-specific data.
We agree that pediatric-specific tools such as the Canadian Paediatric Triage and Acuity Scale (CTAS) or other scoring systems have been proposed, but they are not widely adopted in clinical practice or supported by strong evidence in the context of iGAS. We appreciate the reviewer’s suggestion and have added a comment on this in the revised manuscript (Lines 299-300).
Comments 7: The manuscript has 13 authors, half of the number who answered questions, why don't they evaluate this or that answer? What do they consider correct?
Response 7: We appreciate the reviewer’s comment, but we are not entirely sure we fully understand the question. The authors were involved to varying degrees—some participated in designing the survey, while others contributed by responding to it, depending on their specific expertise.
We chose to include as authors only those individuals who played a substantial role in the conceptualization of the study, data collection, statistical analysis, and drafting or reviewing of the manuscript. Respondents to the survey who did not contribute in these ways were not included as authors.
Comments 8: They could emphasize even more that further work on standardizing the treatment of the disease is essential.
Response 8: We improved this part in the conclusion section (Lines 414-418).
Reviewer 3 Report
Comments and Suggestions for Authors
This is an interesting survey-based study on the management of invasive Group A Streptococcus (iGAS) infections in children. I hope the content of this manuscript will be further improved by addressing the following points:
Abstract
-
The results section states that many practices are commonly shared among clinicians, particularly regarding first-line therapies. I recommend specifying which first-line therapies were agreed upon in detail.
-
The conclusions are currently abstract. Please provide more concrete recommendations regarding the management of iGAS infections in children that can be derived from the results of this study.
Introduction
-
The manuscript cites a 2005 global review estimating 163,000 annual deaths due to iGAS [1]. What is the mortality rate of invasive GAS infections? Please clarify.
-
Please add the risk factors for morbidity and mortality associated with iGAS infections.
-
The manuscript states that between 2011 and 2019, the US Centers for Disease Control and Prevention (CDC) reported increasing clindamycin resistance among iGAS isolates [4]. Could you indicate the specific percentage increase?
-
It is noted that no clear guidelines exist for post-exposure prophylaxis and that recommendations vary worldwide. Please provide representative examples from several countries.
Methods
-
Who developed the survey items, and how were they designed? How was the content reviewed and validated? Please provide more details, including the number of researchers involved and the methods used.
Results
-
Did the participating specialists share any characteristics in terms of the size or geographical location of their institutions?
-
Regarding combined antibiotic therapy (β-lactam plus another GAS-active agent), which other agents were included?
-
Apart from the response that combination therapy should be continued until clinical and hemodynamic stability, what other durations for continuing combination therapy were commonly selected?
-
What evidence and mechanism of action support the addition of linezolid? Please provide references. Also, are there any countries or regions where the addition of linezolid is more commonly practiced?
-
What might explain the differences in agreement rates for first-line therapies depending on the specific disease?
Discussion
-
The manuscript mentions that over the past 20 years, mutations in PBPs leading to increased MICs for penicillin and cephalosporins have been reported in Asia, Africa, and Central America [10]. What changes in prescribing practices have been reported in these regions as a result?
-
As for combination therapy, why might linezolid have been preferred for brain abscesses? Please discuss possible reasons.
Conclusion
-
The conclusion remains somewhat general and abstract. Please add recommendations that are specific and unique to this study.
References
-
The number of cited references is relatively small, and many are rather old. Please incorporate more recent literature to strengthen the manuscript.
Author Response
|
This is an interesting survey-based study on the management of invasive Group A Streptococcus (iGAS) infections in children. I hope the content of this manuscript will be further improved by addressing the following points: Comments 1: Abstract The results section states that many practices are commonly shared among clinicians, particularly regarding first-line therapies. I recommend specifying which first-line therapies were agreed upon in detail. Response 1a: We appreciate the reviewer’s comment,. We specified the first-line therapies with higher agreement among participants in the Abstract results (Lines 42-43). The conclusions are currently abstract. Please provide more concrete recommendations regarding the management of iGAS infections in children that can be derived from the results of this study. |
Response 1b: To provide clinical recommendations is beyond the scope of the present study. Anyway, a more detailed mention on the key strategies to overcome the lack of standardization resulted from the survey was added to the conclusion section of the abstract (Lines 52-54).
Comments 2: Introduction
The manuscript cites a 2005 global review estimating 163,000 annual deaths due to iGAS [1]. What is the mortality rate of invasive GAS infections? Please clarify.
Please add the risk factors for morbidity and mortality associated with iGAS infections.
The manuscript states that between 2011 and 2019, the US Centers for Disease Control and Prevention (CDC) reported increasing clindamycin resistance among iGAS isolates [4]. Could you indicate the specific percentage increase?
Response 2a: We thank the Reviewer for raising this important points. We improved the Introduction section by adding the info required (Lines 68-69, 71-74, 84).
It is noted that no clear guidelines exist for post-exposure prophylaxis and that recommendations vary worldwide. Please provide representative examples from several countries.
Response 2b: We added examples of the available guidelines in the Discussion section (Lines 320-335).
Comments 3: Introduction
Who developed the survey items, and how were they designed? How was the content reviewed and validated? Please provide more details, including the number of researchers involved and the methods used.
Response 3: Initially, a comprehensive literature review was conducted to identify well-established areas and key uncertainties regarding iGAS management. Based on this, two researchers developed clinical questions using the PICO framework (Patient/Population/Problem, Intervention, Comparison, Outcome). Each component of the PICO questions was then carefully translated into clear, specific survey items designed to be easily understood and answered by the target audience, minimizing ambiguity. To ensure methodological rigor and transparency, the CROSS checklist was applied to address essential survey elements. Following this, the survey draft was submitted to the SITIP Council for review, feedback, and approval prior to its distribution among PID specialists.
A sentence was added in the Methods section to better explain this process (Lines 391-403).
Comments 4: Results
Did the participating specialists share any characteristics in terms of the size or geographical location of their institutions?
Response 4a: The participating specialists were primarily based in tertiary care hospitals across northern, central, and southern regions of Italy. European participants mainly came from Southern European countries. While the size of the institutions varied, all were tertiary care centers with dedicated personnel specialized in the management of pediatric infectious diseases.
Regarding combined antibiotic therapy (β-lactam plus another GAS-active agent), which other agents were included?
Response 4b: The antibiotic agents considered for combination therapy with a β-lactam are presented in Figure 1, panel C. These include clindamycin, linezolid, tedizolid, and vancomycin, with the corresponding levels of agreement among respondents indicated in the figure.
Apart from the response that combination therapy should be continued until clinical and hemodynamic stability, what other durations for continuing combination therapy were commonly selected?
Response 4c: We apologize, as there was likely an issue with uploading Table 1 (Supplementary file) to the platform. The table contains the percentages of responses for each option of the question. The Table should now be available.
What evidence and mechanism of action support the addition of linezolid? Please provide references. Also, are there any countries or regions where the addition of linezolid is more commonly practiced?
Response 4d: Data on the use of linezolid in pediatric iGAS infections are limited. Similar to clindamycin, linezolid is a protein synthesis inhibitor and may play a role in disrupting the production of virulence factors such as streptolysin S, protein M, and pyrogenic exotoxins. Adult studies suggest that linezolid may be non-inferior to clindamycin in terms of clinical outcomes, with additional benefits including a lower risk of Clostridium difficile infection and effective coverage against methicillin-resistant Staphylococcus aureus (MRSA) (Babiker A et al, Lancet Infect Dis 2025;25:265-75). Its excellent pulmonary penetration may be particularly advantageous in cases of iGAS pneumonia. Recently, Buricchi L et al. reported a cohort of pediatric patients in which linezolid was effective in refractory cases unresponsive to the beta-lactam plus clindamycin combination therapy. Relevant references have been added to the manuscript.
Regarding geographic use, evidence on linezolid use in pediatric patients across Europe is lacking. However, it’s important to note that its use in pediatrics still raises some concerns due to limited safety data and its off-label status in several European countries, including Italy. We added a sentence in the text to specify this consideration (Lines 323-324).
What might explain the differences in agreement rates for first-line therapies depending on the specific disease?
Response 4e: Thank you for this interesting observation. We believe that the differences in agreement rates regarding first-line therapies may be influenced by several factors, including the clinicians’ expertise, the availability of antibiotics at each hospital, and local prescribing policies. Some institutions may prefer to start with broader-spectrum agents followed by de-escalation once microbiological results are available, while others may opt for a narrow-spectrum antibiotic initially and escalate treatment if needed. These variations likely contribute to the observed discrepancies in treatment preferences across specific diseases.
Comments 5: Discussion
The manuscript mentions that over the past 20 years, mutations in PBPs leading to increased MICs for penicillin and cephalosporins have been reported in Asia, Africa, and Central America [10]. What changes in prescribing practices have been reported in these regions as a result?
Response 5a: We did not find robust evidence in the literature describing specific changes in antibiotic prescribing practices in Asia, Africa, or Central America following the emergence of PBP mutations and increased MICs to penicillin and cephalosporins. To our knowledge, no regional or national guidelines have been formally updated in response to these findings.
A recent study by Gregory C. et al. (JAMA, 2025), which analyzed the incidence of iGAS infections across 10 U.S. states, highlights the need for further evaluation of the clinical impact of reduced clindamycin susceptibility but does not report on specific changes in prescribing patterns. This underscores the broader lack of data on how emerging resistance is influencing clinical practice globally.
As for combination therapy, why might linezolid have been preferred for brain abscesses? Please discuss possible reasons.
Response 5b: Several studies have described a better tissue permeability of linezolid, including across the blood–brain barrier, compared to vancomycin and clindamycin [Sullins A et al. Paediatr Drugs. 2013 Apr;15(2):93-117. doi: 10.1007/s40272-013-0017-5. Pharmacokinetics of antibacterial agents in the CSF of children and adolescents]. Additionally, linezolid inhibits the initiation phase of bacterial protein synthesis with effects on toxin production (references provided before).
Comments 6: Conclusion
The conclusion remains somewhat general and abstract. Please add recommendations that are specific and unique to this study.
Response 6: We appreciate the reviewer’s suggestion. As mentioned in Response 1a, providing specific clinical recommendations is beyond the scope of the present study, which primarily aims to describe current practices and identify areas of uncertainty. However, we have revised the conclusions to more clearly emphasize the urgent need for further research to standardize the management of pediatric iGAS infections and to develop evidence-based guidelines tailored to this population (Lines 414-418).
Comments 7: References
The number of cited references is relatively small, and many are rather old. Please incorporate more recent literature to strengthen the manuscript.
Response 7: Recent references has been added.
Round 2
Reviewer 1 Report
Comments and Suggestions for Authors
You have done your job.
Comments 5: Only 54.1% agreement, What do you believe is driving this lack of consensus, and what does it suggest about the current state of evidence-based guidelines.
Response 5: It is not possible to answer this question. Please indicate which lines in the manuscript you are referring to.
It's line 107, results section sepsis
Author Response
Comments 5: Only 54.1% agreement, What do you believe is driving this lack of consensus, and what does it suggest about the current state of evidence-based guidelines. It's line 107, results section sepsis.
Response 5: Thank you for your clarification. The relative lack of consensus regarding the choice of ceftriaxone over penicillin or ampicillin as first-line therapy for sepsis is likely attributable to the fact that the question did not specifically address the selection of an empirical regimen. According to the latest guidelines on pediatric sepsis management (Surviving Sepsis Campaign 2020), a 3rd generation cephalosporin would represent the preferred first-line option for establishing an empiric broad-spectrum antimicrobial therapy. However, in the context of narrowing the spectrum relying on microbiological evidence, penicillins appear to be the most reasonable choice. As already emphasized in other parts of the manuscript, any evidence-based guideline is currently available on this matter. Consequently, we believe the choice to prefer ceftriaxon or penicillins as targeted therapy can only be guided by the personal judgment of the clinicians, in line with their awareness of antimicrobial stewardship principles.
Reviewer 3 Report
Comments and Suggestions for Authors
The revisions made adequately address the points raised, and I believe the content is worthy of publication.
Author Response
Comments 1: The revisions made adequately address the points raised, and I believe the content is worthy of publication.
Response 1: Thank you for your positive appraisal of the revised manuscript.